# Preconception Lifestyle and Cardiovascular Health in the Offspring of Overweight and Obese Women

**DOI:** 10.3390/nu11102446

**Published:** 2019-10-14

**Authors:** Tessa M. van Elten, Cornelieke van de Beek, Anouk Geelen, Reinoud J.B.J. Gemke, Henk Groen, Annemieke Hoek, Ben Willem Mol, Mireille N.M. van Poppel, Tessa J. Roseboom

**Affiliations:** 1Amsterdam UMC, Department of Public and Occupational Health, Vrije Universiteit Amsterdam, De Boelelaan 1117, 1105 AZ Amsterdam, The Netherlands; mireille.van-poppel@uni-graz.at; 2Amsterdam UMC, Department of Clinical Epidemiology, University of Amsterdam, Biostatistics and Bioinformatics, Meibergdreef 9, 1105 AZ Amsterdam, The Netherlands; 3Amsterdam UMC, Department of Obstetrics and Gynecology, University of Amsterdam, Meibergdreef 9, 1105 AZ Amsterdam, The Netherlands; 4Amsterdam Public Health Research Institute, 1105 AZ Amsterdam, The Netherlands; 5Amsterdam Reproduction and Development, 1105 AZ Amsterdam, The Netherlands; 6Division of Human Nutrition, Wageningen University & Research, 6700 AA Wageningen, The Netherlands; anouk.geelen@wur.nl; 7Department of Pediatrics, Emma Children’s Hospital, Vrije Universiteit Amsterdam, Amsterdam UMC, 1081 HV Amsterdam, The Netherlands; 8Department of Epidemiology, University of Groningen, University Medical Centre Groningen, 9700 RB Groningen, the Netherlands; h.groen01@umcg.nl; 9Department of Obstetrics and Gynecology, University of Groningen, University Medical Centre, Groningen, 9700 RB Groningen, the Netherlands; a.hoek@umcg.nl; 10Department of Obstetrics and Gynecology, Monash University, 3800 Melbourne, Australia; b.w.mol@amc.uva.nl; 11Institute of Sport Science, University of Graz, 8010 Graz, Austria

**Keywords:** preconception dietary intake, preconception physical activity, offspring anthropometry, offspring blood pressure, offspring pulse wave velocity

## Abstract

Women’s lifestyle has important implications for the development and health of their offspring. Yet little is known about the association between women’s preconception dietary intake and physical activity with cardiovascular health of the offspring. We therefore examined this association in a group of Dutch women with overweight or obesity (BMI ≥ 29 kg/m^2^) and infertility, who participated in a 6-month randomized preconception lifestyle intervention trial, and their offspring (*n* = 46). Preconception dietary intake and physical activity were assessed during the 6-month intervention using a food frequency questionnaire and the Short QUestionnaire to ASsess Health-enhancing physical activity (SQUASH), respectively. Offspring cardiovascular health (i.e., BMI, waist:height ratio, systolic and diastolic blood pressure, fat and fat free mass, and pulse wave velocity) was measured at age 3–6 years. Multivariable linear regression analyses were used to examine the associations between preconception lifestyle and offspring cardiovascular health. Higher preconception vegetable intake (per 10 g/day) was associated with lower offspring diastolic blood pressure (Z-score: −0.05 (−0.08; −0.01); *p* = 0.007) and higher preconception fruit intake (per 10 g/day) was associated with lower offspring pulse wave velocity (−0.05 m/s (−0.10; −0.01); *p* = 0.03). Against our expectations, higher preconception intake of sugary drinks was associated with a higher offspring fat free mass (0.54 kg (0.01; 1.07); *p* = 0.045). To conclude, preconception dietary intake is associated with offspring health.

## 1. Introduction

Women’s lifestyle has important implications for the growth, development and health of their offspring [1,2,3]. Suboptimal conditions during the earliest stages of embryonic and fetal development have lasting consequences for cardiovascular and metabolic health, and thereby increase an individual’s risk of cardiovascular disease in later life [4]. This phenomenon is known as developmental programming. 

However, lifestyle interventions in pregnancy did not show the desired improvements in offspring health outcomes, suggesting that these lifestyle interventions were initiated after the most critical period of early embryo and fetal development [5]. Therefore, interventions starting during the preconception period might have a larger effect on offspring health [1]. Yet little is known about the association between women’s preconception dietary intake and physical activity, and cardiovascular health of the offspring [2]. 

The LIFEstyle study was a large randomized controlled trial (RCT) studying the effects of a preconception lifestyle intervention in overweight and obese, infertile women [6,7]. The 6-month preconception lifestyle intervention favorably changed lifestyle during the intervention [8], leading to reduced weight [7], and improved cardiometabolic health at the end of the intervention period [9]. Follow-up of the children at age 3–5 years did not show differences in cardiovascular health between children of mothers in the intervention and control group, possibly due to limited statistical power and the large inter individual responses to the intervention (Mintjens under review). Throughout the intervention, women filled out questionnaires regarding their dietary intake and physical activity, giving us the unique opportunity to study the link between lifestyle before conception and cardiovascular health outcomes in the offspring.

We hypothesized that women with a more healthy diet, i.e., higher vegetable and fruit consumption and lower intake of high caloric snacks and beverages, and with more moderate to vigorous physical activity (MVPA) before conception, have offspring with a lower BMI, lower waist:height ratio, lower blood pressure, lower fat mass and higher fat free mass, and lower pulse wave velocity (PWV). Since the literature suggests a positive effect of restoring the energy balance by increasing physical activity [10], we additionally examined the combined effect of preconception dietary intake and physical activity on cardiovascular health of the offspring.

## 2. Materials and Methods

### 2.1. Study Population

The LIFEstyle study was a large multicenter RCT conducted between 2009 and 2014 in the Netherlands (Dutch trial register; NTR 1530). The design and results of the LIFEstyle study are described in detail previously [6,7]. In brief, the original study population consisted of 577 infertile women between 18 and 39 years old, with a BMI of ≥29 kg/m^2^. Participants were randomized into a 6-month structured lifestyle intervention program (intervention group), or promptly started infertility care as usual (control group). The intervention was based on the Dutch dietary guidelines of 2006 [11] and the Dutch physical activity guidelines [12]. All singleton children, conceived within 24 months after randomization of their mothers into the LIFEstyle study, were eligible to participate. The follow-up study (WOMB project) was conducted in 2016 and 2017 when the children were aged 3–6 years old [13]. 

The LIFEstyle study as well as the WOMB project were conducted according to the guidelines laid down in the Declaration of Helsinki and all procedures were approved by the Medical Ethics Committee of the University Medical Centre Groningen, the Netherlands (METc 2008/284). Written informed consent was obtained from all female participants at the start of the LIFEstyle study as well as from both parents for the follow-up study. Although the original trial was set up to assess effects on live birth and hence the sample size was determined based on this outcome, post-hoc power calculations were done to assess statistical power to detect differences in child health revealing that with a drop-out rate of 50%, the study would have 80% power to detect a 0.3 unit difference in BMI between children of mothers in the intervention group and children in the control group (with an SD of 1.5 and at an alpha level of 0.05: a reduction in BMI from 16 to 15.7). 

### 2.2. Women’s Preconception Diet and Physical Activity

During participation in the LIFEstyle study, all women were asked to fill out a 33-item food frequency questionnaire (FFQ) and the Short QUestionnaire to ASsess Health-enhancing physical activity (SQUASH) four times: at the start of the intervention and at 3, 6, and 12 months after randomization into the trial. 

The FFQ was based on the standardized questionnaire on food consumption used for the Public Health Monitor in the Netherlands [14], supplemented with questions about the intake of snacks, soda and alcoholic beverages. Data were collected about frequency of consumption per week or per month. Portion size was asked per standard household measure. Preconception dietary intake was operationalized as the last reported intake of vegetables (raw as well as cooked; in units of 10 g/day), fruit (10 g/day), sugary drinks (fruit juice and soda; glasses/day), savory snacks (crisps, pretzels, nuts and peanuts; handful/week) and sweet snacks (biscuits, pieces of chocolate, candies or liquorices; portion/week) before the start of the pregnancy. Additionally, we examined preconception diet using a self-composed composite score. All five food groups were split based on median intake of the total study population. For vegetables and fruit, an intake equal to or below the median scored zero points and an intake above the median scored one point. For sugary drinks, savory and sweet snacks, the scoring was reversed. The food composite score ranges between 0 and 5 points; the higher the score, the healthier the dietary intake. 

The validated SQUASH questionnaire [15] was used to collect information about commuting activities, leisure time activities, household activities, and activities at work and school, using three main questions: days per week active, average time per day/week spending in that particular activity (hours and/or minutes), and intensity of the activity (low, moderate, high). Preconception physical activity was operationalized as the last reported total MVPA (hour/week) measurement before the start of the pregnancy.

### 2.3. Offspring Cardiovascular Health

Offspring cardiovascular health was measured during the WOMB project by physical examinations in a mobile research vehicle, which ensured a standardized environment. Examinations were done by trained researchers, according to a standardized research protocol. Children were asked not to eat or drink from 90 min before the mobile research vehicle arrived at their home and to empty their bladder before the measurements started. Height was measured on bare feet to the nearest 0.1 cm using a wall stadiometer (SECA 206; SECA, Germany). Weight was measured in underwear to the nearest 0.1 kg using a digital weighting scale (SECA 877; SECA, Germany), while the child was standing still and looking straight ahead. BMI was calculated by dividing the weight of the child in kilograms by their height in meters squared. Waist circumference was measured to the nearest 0.1 cm using a flexible measurement tape (SECA 201; SECA, Germany). Waist to height ratio was calculated by dividing the waist circumference in centimeters by the height of the child in centimeters. All anthropometric measurements were done twice, and in case of >0.5 cm difference in height, >0.5 kg difference in weight, and >1 cm difference in waist circumference between the two measurements, a third measurement was done. After the child sat quietly for 5 min, blood pressure was measured three times at heart level, in sitting position at the non-dominant arm, using an automatic measurement device (Omron HBP-1300; OMRON Healthcare, The Netherlands) with appropriate cuff size. The child was not allowed to talk in between. Time in between the subsequent measurements was 30 s. After the child laid quietly for 5 min, body composition was measured by bio-electrical impedance (BIA; Bodystat 1500; Bodystat Ltd., Isle of Man, UK). Electrode strips were attached at the dorsal side of the left hand and foot, with at least 3–5 cm in between the two electrodes. If the impedance or resistance differed >5 Ω a third measurement was done. The recalibrated Kushner equation [16] was used to estimate total body water, after which fat free mass (kg) and fat mass (as percentage of total body weight) were calculated. Directly after the BIA measurement, still in supine position, carotid-femoral PWV was measured twice using the Complior Analyse (Complior; Alam medical, Saint-Quentin-Fallavier, France). Mechanotransducer sensors were placed at the carotid artery on the right side and on the femoral artery on the left side. Blood pressure in laying position was measured once before the actual measurement started and entered in the Complior software. Directly after the measurement, distance between both sensors was measured and also entered in the Complior software. If the results of the two measurements differed >10% from each other a third measurement was done. For all physical examinations, mean values of the two or three measurements were used for data analysis.

### 2.4. Covariates

The following covariates were obtained from questionnaires filled out by the mother: women’s ethnicity (Caucasian or not), women’s education level at randomization (no education or primary school, secondary education, intermediate vocational education, higher vocational education and university), smoking at randomization (yes/no), and offspring breastfeeding (months exclusively breastfed). Time in between the last measurement of women’s preconception lifestyle and conception date (weeks), women’s age at the measurement of preconception lifestyle (years), and age of the child at the time of the physical examinations (years) were calculated. Women’s diagnosis with polycystic ovary syndrome (yes/no), offspring sex (boy/girl), birth weight (grams), and mode of conception (spontaneous, ovulation induction, intra uterine insemination, in vitro fertilization/intracytoplasmic sperm injection/cryotherapy) were obtained from medical records. To correct for current diet and physical activity of the offspring, we added total energy intake (kcal) and physical activity as counts per minute (counts/min) to our models. Offspring dietary intake was recorded by their parents/caregivers using a standardized food record for three non-consecutive days, including one weekend day. Data on offspring physical activity was measured for seven consecutive days using the triaxial Actigraph wGT3X-BT or GT3X+ monitor. We additionally examined if women’s allocated randomization group during the LIFEstyle study (intervention or control group) affected the associations by adding a dichotomous variable for treatment allocation to the model.

### 2.5. Data Analysis

Linear regression analysis was used to examine the association between preconception diet and physical activity and cardiovascular health of the offspring. Residuals were shown to be normally distributed. Results are displayed as regression coefficients and 95% confidence intervals (C.I.). We additionally combined preconception diet, as the self-composed composite score, and preconception MVPA into one regression model and studied the interaction between those two independent variables by adding an interaction term into the model. Due to extreme values (+/− 3 SD), we excluded five woman-child pairs from our analyses regarding preconception MVPA and offspring cardiovascular health (>33 h preconception MVPA per week). 

For the cardiovascular outcomes, age and sex specific BMI Z-scores were calculated by the lambda-mu-sigma (LMS) method using the World Health Organization (WHO) BMI growth standards [17]. And age and sex specific blood pressure Z-scores were calculated based on the fourth report on the diagnosis, evaluation, and treatment of high blood pressure in children and adolescents [18]. Based on biological implausibility, we excluded two children with a body fat percentage of <5% from our statistical analysis regarding body composition. We checked if offspring sex was an effect modifier by adding an interaction term into the regression models. Covariates were added one by one into the univariate regression models. Effect estimates were small and hence adding covariates into the univariate model changed the majority of the effect estimates by 10% or more. We therefore decided only to add covariates into our model if our conclusions changed based on the adjusted effect estimates and 95% C.I. 

Statistical analyses were performed using the software Statistical Package for the Social Sciences (SPSS) version 24 for Windows (SPSS, Chicago, IL, USA). *p*-values < 0.05 were considered statistically significant. 

## 3. Results

In total, 341 children were conceived within 24 months after the start of the LIFEstyle study and 305 singletons were eligible to participate. We received informed consent of the parents of 51 children (16.7%) (Figure 1). 

Table 1 shows the characteristics of the participating women and their offspring (*n* = 46). 

At follow-up, offspring had a mean BMI of 16.5 kg/m^2^ (SD: 1.7) (Z-score 0.69 (SD: 1.05)), with a fat mass of 21.0% (SD: 8.7) (Table 2). We observed no statistically significant differences comparing the women and offspring characteristics of participants versus non-participants (Appendix A).

Higher preconception vegetable intake (per 10 g/day) was associated with a lower diastolic blood pressure (DBP) in the offspring (Z-score: −0.05 (−0.08; −0.01); *p* = 0.007) (Table 3). Higher preconception fruit intake (10 g/day) was associated with a lower PWV in the offspring (−0.05 m/s (−0.10; −0.01); *p* = 0.03). Higher intake of preconception sugary drinks (glass/day) was associated with a higher offspring fat free mass in the adjusted model (unadjusted: 0.56 kg (−0.14; 1.27); *p* = 0.11; adjusted: 0.54 kg (0.01; 1.07); *p* = 0.045). We did not observe any other statistically significant associations between preconception diet and physical activity and offspring cardiovascular health. Preconception total food score was not statistically significantly associated with offspring cardiovascular health (Appendix A). 

We did not observe any statistically significant associations when preconception food score and preconception MVPA were both added into the model (Table 4). 

Additionally, there was no statistically significant interaction between both independent variables for any of the offspring outcomes (results not shown). Furthermore, we did not observe any effect modification by offspring sex, nor did our conclusions change when adding women related covariates (ethnicity, education level, age at the last preconception measurement, mode of conception, time in between the last measurement of preconception lifestyle and conception date, randomization group, PCOS, pregnancy BMI, and smoking) and offspring related covariates (sex, age at time of the physical examinations, breastfeeding, birth weight, current dietary intake (kcal) and physical activity (counts/min)) into our models.

## 4. Discussion

Higher intake of vegetables before conception was associated with lower offspring DBP, and higher intake of fruit before conception was associated with lower offspring PWV. We additionally observed an unexpected association between higher intake of sugary drinks before conception and higher offspring fat free mass. 

### 4.1. Interpretation of Results

Due to the relatively small number of mother-child pairs in the study, the effect estimates have wide confidence intervals and the clinical relevance of these findings are yet to be determined. The association of higher preconception vegetable intake with lower offspring DBP is in line with a previous study showing that the higher the Mediterranean Diet Score (MDS) during pregnancy—which is characterized by a high intake of vegetables—the lower the offspring DBP (3-point increment in MSD: −0.57 mmHg (−0.98; −0.16)) [19]. Another study, however, did not find that higher vegetable intake in late pregnancy was associated with lower offspring DBP [20]. It might be that vegetable intake during the preconception period has a larger influence on offspring DBP compared to vegetable intake in late pregnancy, as the fetal heart and organs develop very early in pregnancy and beneficial effects of vegetable intake preconceptionally can optimally affect cardiovascular development. 

There are several underlying mechanisms that might explain the observed associations of preconception vegetable and fruit intake with offspring’s cardiovascular health. Fruit and vegetable intake is inversely associated with CRP concentrations [21] and higher maternal CRP levels are associated with childhood adiposity [22]. Furthermore, it might be that a higher intake of antioxidants, when consuming a diet rich in fruit and vegetables, is causing the more favorable cardiovascular health in the offspring [23]. 

Our finding of increased sugary drinks before conception being associated with higher offspring fat free mass is not in line with literature [24,25,26]. It might be that the category sugary drinks in our FFQ is too broad, including both artificial sweetened beverages and fresh juices. Fruit and vegetable juices seem to favorably affect women’s cardiometabolic health [27], which might explain the offspring’s more favorable body composition. However, Jen and colleagues (2017) showed that fruit juice intake (100% fruit juice only) during pregnancy was associated with a higher offspring fat mass. Given the number of associations studied we cannot exclude the possibility that this is a chance finding.

This study is unique with respect to the preconception time window during which dietary intake and physical activity were measured. Our findings add to the current literature that is largely limited to lifestyle during pregnancy. We did not examine women’s lifestyle during pregnancy, but literature shows that women increase their fruit and vegetable intake and decrease their intake of fried and fast food, coffee, and tea during pregnancy [28], which may interact with the effects of preconception intake on offspring health. Future studies should therefore take both preconception and pregnancy intakes into account to assess the combined effects on offspring health.

### 4.2. Strengths and Limitations

An important strength of the current study is the reliable and extensive data collection regarding cardiovascular health of the children. Additionally, we studied the association of both preconception dietary intake and physical activity with offspring cardiovascular health, to test whether these two lifestyle behaviors influenced each other in their association with offspring cardiovascular health. We collected detailed information about offspring current lifestyle, enabling us to study if associations between preconception lifestyle and offspring cardiovascular health were confounded by offspring current lifestyle, which was not the case. This suggests that offspring vascular health is programmed by preconception lifestyle independently of current offspring diet and physical activity. 

A major limitation of the current study is the small sample size, hence it is not possible to conclude if there truly are no associations between certain preconception lifestyle factors and offspring health, or that we were not able to observe them. Moreover, because of the small sample size, extreme values in our data had a large impact on the associations. We studied a large number of associations and were therefore not able to exclude the possibility of chance findings. Another limitation is the use of self-reported preconception lifestyle instead of objective measurements. Obese women tend to underreport unhealthy lifestyle behaviors and over-report healthy lifestyle behaviors [29], which could have biased our results. Finally, we used a short FFQ to measure women’s dietary intake. Literature suggests that the complete dietary pattern is involved in developmental programming [30]. Nevertheless, we speculate that the food groups included in our study fairly represent eating a healthy diet, which is high in fruit and vegetables and low in sugary drinks and snacks.

### 4.3. Generalizability and Recommendations for Future Research

Our study population included children of women who were infertile. Offspring of infertile women have a higher diastolic blood pressure compared to offspring of fertile women, irrespective of maternal BMI before pregnancy [31]. However, our findings cannot be explained by differences in dietary intake between women who conceived spontaneously and through artificial reproductive technologies, as correction for mode of conception (spontaneous, ovulation induction, IUI, IVF/ICSI/CRYO) did not affect the associations. Since infertility is associated with stress [32], which influences lifestyle [33] as well as offspring health [34], future research should examine if our results are generalizable to all women of childbearing age. Our sample size was small and therefore preconception research in larger study populations should replicate our findings.

## 5. Conclusions

Preconception dietary intake is associated with offspring health. Higher intakes of vegetable and fruit before conception were associated with better cardiovascular health in the offspring.

## Figures and Tables

**Figure 1 nutrients-11-02446-f001:**
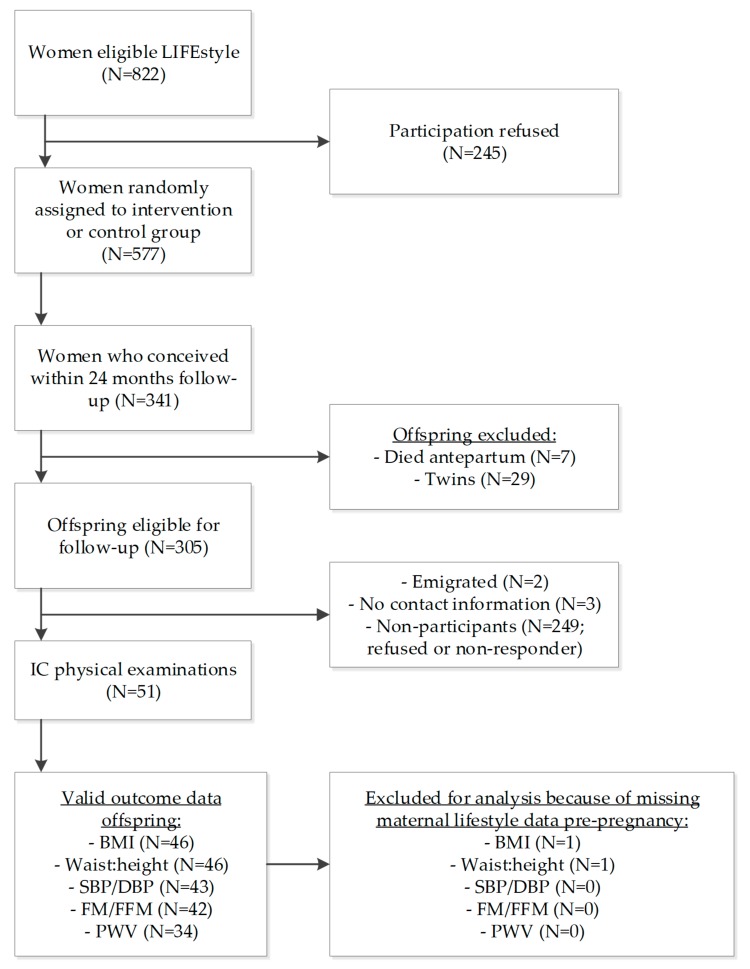
Flow-chart of the women and their offspring included in this study; IC = informed consent. Total numbers of participants used for analysis differ per dependent/independent variable based on available data.

**Table 1 nutrients-11-02446-t001:** Study characteristics of the participating women and their offspring (*n* = 46) ^1^.

Women’s Characteristics	
**Age at Pregnancy (Years; Mean; SD)**	**30.1 (3.9)**
Caucasian (yes; *n*; %)	44 (95.7)
Education level (*n*; %)	
No education or primary school	0
Secondary school	10 (22.7)
Intermediate vocational education	26 (59.1)
Higher vocational education and university	8 (18.2)
Nulliparous (yes; *n*; %)	34 (73.9)
Pre-pregnancy BMI (kg/m^2^; mean; SD)	35.3 (3.4)
Gestational diabetes (yes; *n*; %)	11 (23.9)
Smoking at randomization (yes; *n*; %)	8 (17.4)
Mode of conception (*n*; %)	
Spontaneous	18 (39.1)
Ovulation induction	16 (34.8)
Intra Uterine Insemination	7 (15.2)
IVF/ICSI/CRYO	5 (10.9)
PCOS (yes; *n*; %)	20 (43.5)
**Offspring’s Characteristics**	
Sex (boys; *n*; %)	22 (47.8)
Birth weight (grams; mean; SD)	3497 (507)
Gestational age at birth (weeks; mean; SD)	39.1 (1.7)
Exclusively breastfed (months; median; IQR)	0.0 (0.0; 2.0)
Age of the child at time of physical examinations (years; mean; SD)	4.7 (1.0)

^1^ Missing data for education level: *n* = 2 and breastfeeding: *n* = 4. SD = standard deviation; BMI = Body Mass Index; IVF = In Vitro Fertilization; ICSI = Intracytoplasmic Sperm Injection; CRYO = Cryotherapy; PCOS = polycystic ovary syndrome.

**Table 2 nutrients-11-02446-t002:** Offspring cardiovascular health outcomes at the ages 3–6 years old.

Offspring Health	*n*	Mean; SD
BMI (kg/m^2^)	46	16.5 (1.7)
BMI (Z-Score) ^1^	46	0.69 (1.05)
Waist to height ratio (waist/height)	46	0.49 (0.03)
Systolic blood pressure (mmHg)	43	100.2 (7.5)
Diastolic blood pressure (mmHg)	43	64.2 (7.0)
Systolic blood pressure (Z-Score) ^2^	43	0.51 (0.59)
Diastolic blood pressure (Z-Score) ^2^	43	0.94 (0.59)
Fat mass (percentage)	42	21.0 (8.7)
Fat free mass (kg)	42	15.6 (2.3)
Pulse wave velocity (m/s)	34	4.5 (1.0)

^1^ Z-Scores BMI were calculated using the WHO BMI growth standards, LMS method. ^2^ Z-Scores blood pressure were calculated based on “The fourth report on the diagnosis, evaluation, and treatment of high blood pressure in children and adolescents” of the National Heart, Lung and Blood institute.

**Table 3 nutrients-11-02446-t003:** Linear regression analyses of preconception diet and physical activity and offspring cardiovascular health at age 3–6 years.

**Preconception Vegetable Intake (10 g/day)**					
Offspring health	*n*	β (95% C.I.) unadjusted	*p*-value	β (95% C.I.) adjusted ^1^	*p*-value
BMI (Z-Score)	45	−0.01 (−0.07; 0.06)	0.83	NA	NA
Waist:height (ratio)	45	0.00 (−0.002; 0.002)	0.95	0.00 (−0.001; 0.002)	0.82
SBP (Z-Score)	43	0.001 (−0.04; 0.04)	0.95	NA	NA
DBP (Z-Score)	43	−0.05 (−0.08; −0.01)	0.007	NA	NA
Fat mass (%)	42	0.04 (−0.50; 0.59)	0.87	0.03 (−0.47; 0.53)	0.91
Fat free mass (kg)	42	0.07 (−0.08; 0.21)	0.34	0.02 (−0.08; 0.13)	0.66
PWV (m/s)	34	−0.03 (−0.10; 0.05)	0.49	−0.05 (−0.13; 0.03)	0.19
**Preconception Fruit Intake (10 g/day)**					
Offspring health	*n*	β (95% C.I.) unadjusted	*p*-value	β (95% C.I.) adjusted ^1^	*p*-value
BMI (Z-Score)	45	0.01 (−0.04; 0.05)	0.76	NA	NA
Waist:height (ratio)	45	0.00 (−0.001; 0.001)	0.90	0.00 (−0.001; 0.001)	0.65
SBP (Z-Score)	43	−0.01 (−0.03; 0.02)	0.64	NA	NA
DBP (Z-Score)	43	−0.02 (−0.04; 0.01)	0.17	NA	NA
Fat mass (%)	42	0.08 (−0.26; 0.43)	0.63	0.09 (−0.24; 0.42)	0.60
Fat free mass (kg)	42	−0.03 (−0.12; 0.06)	0.51	0.01 (−0.06; 0.08)	0.69
PWV (m/s)	34	−0.06 (−0.10; −0.01)	0.02	−0.05 (−0.10; −0.01)	0.03
**Preconception Sugary Drink Intake (Glass/Day)**					
Offspring health	*n*	β (95% C.I.) unadjusted	*p*-value	β (95% C.I.) adjusted ^1^	*p*-value
BMI (Z-Score)	43	0.02 (−0.29; 0.32)	0.91	NA	NA
Waist:height (ratio)	43	0.002 (−0.01; 0.01)	0.67	0.002 (−0.01; 0.01)	0.63
SBP (Z-Score)	41	0.07 (−0.10; 0.25)	0.38	NA	NA
DBP (Z-Score)	41	0.12 (−0.05; 0.28)	0.16	NA	NA
Fat mass (%)	40	1.53 (−1.18; 4.24)	0.26	0.65 (−1.94; 3.24)	0.61
Fat free mass (kg)	40	0.56 (−0.14; 1.27)	0.11	0.54 (0.01; 1.07)	0.045
PWV (m/s)	32	−0.06 (−0.41; 0.30)	0.76	−0.06 (−0.44; 0.33)	0.77
**Preconception Savory Snack Intake (Handful/Week)**					
Offspring health	*n*	β (95% C.I.) unadjusted	*p*-value	β (95% C.I.) adjusted ^1^	*p*-value
BMI (Z-Score)	43	0.03 (−0.05; 0.10)	0.45	NA	NA
Waist:height (ratio)	43	0.002 (0.00; 0.004)	0.08	0.001 (0.00; 0.003)	0.12
SBP (Z-Score)	41	0.01 (−0.03; 0.05)	0.74	NA	NA
DBP (Z-Score)	41	−0.01 (−0.05; 0.03)	0.60	NA	NA
Fat mass (%)	40	−0.19 (−0.81; 0.43)	0.54	−0.02 (−0.59; 0.55)	0.93
Fat free mass (kg)	40	−0.02 (−0.19; 0.14)	0.78	0.03 (−0.09; 0.16)	0.57
PWV (m/s)	32	−0.04 (−0.14; 0.05)	0.37	−0.04 (−0.14; 0.06)	0.45
**Preconception Sweet Snack Intake (Portion/Week) ^2^**					
Offspring health	*n*	β (95% C.I.) unadjusted	*p*-value	β (95% C.I.) adjusted ^1^	*p*-value
BMI (Z-Score)	43	−0.01 (−0.09; 0.07)	0.78	NA	NA
Waist:height (ratio)	43	−0.001 (−0.003;0.001)	0.56	−0.001 (−0.003; 0.001)	0.43
SBP (Z-Score)	41	0.02 (−0.03; 0.06)	0.45	NA	NA
DBP (Z-Score)	41	0.03 (−0.01; 0.07)	0.11	NA	NA
Fat mass (%)	40	0.02 (−0.62; 0.66)	0.95	0.08 (−0.50; 0.66)	0.78
Fat free mass (kg)	40	−0.07 (−0.23; 0.10)	0.42	−0.03 (−0.15; 0.10)	0.66
PWV (m/s)	32	−0.01 (−0.09; 0.07)	0.80	0.00 (−0.08; 0.08)	>0.99
**Preconception Total Moderate to Vigorous Physical Activity (Hour/Week)**					
Offspring health	*n*	β (95% C.I.) unadjusted	*p*-value	β (95% C.I.) adjusted ^1^	*p*-value
BMI (Z-Score)	40	0.004 (−0.05; 0.05)	0.87	NA	NA
Waist:height (ratio)	40	0.001 (−0.001; 0.002)	0.27	0.001 (−0.001; 0.002)	0.36
SBP (Z-Score)	38	−0.004 (−0.03; 0.03)	0.81	NA	NA
DBP (Z-Score)	38	−0.01 (−0.04; 0.02)	0.51	NA	NA
Fat mass (%)	37	−0.17 (−0.62; 0.28)	0.45	−0.02 (−0.44; 0.41)	0.93
Fat free mass (kg)	37	−0.02 (−0.15; 0.11)	0.77	0.02 (−0.08; 0.11)	0.71
PWV (m/s)	29	0.01 (−0.04; 0.06)	0.77	0.01 (−0.04; 0.06)	0.66

BMI = body mass index; SBP = systolic blood pressure; DBP = diastolic blood pressure; PWV = pulse wave velocity; NA = not applicable. ^1^ Adjusted for offspring sex and offspring age. We did not adjust outcomes expressed in Z-scores, as sex and age were already taken into account when calculating a Z-score. ^2^ One portion of sweet snacks included two biscuits, two pieces of chocolate, five candies, or five pieces of liquorice.

**Table 4 nutrients-11-02446-t004:** Linear regression analysis of the preconception food score (0–5 points; the higher the more healthy) and preconception moderate to vigorous physical activity (MVPA) and offspring cardiovascular health at age 3–6 years ^1^.

Preconception Food Score (0–5 Points) and Preconception MVPA (Hour/Week)									
		β (95% C.I.) Unadjusted				β (95% C.I.) Adjusted ^2^			
	*n*	Food score	*p*	MVPA	*p*	Food score	*p*	MVPA	*p*
BMI (Z-Score)	39	−0.03 (−0.33; 0.26)	0.82	−0.001 (−0.06; 0.06)	0.97	NA	NA	NA	NA
Waist:height (ratio)	39	−0.002 (−0.01; 0.01)	0.56	0.00 (−0.001; 0.002)	0.53	−0.003 (−0.01; 0.01)	0.45	0.00 (−0.001; 0.002)	0.52
SBP (Z-Score)	37	−0.03 (−0.20; 0.13)	0.67	0.01 (−0.02; 0.04)	0.59	NA	NA	NA	NA
DBP (Z-Score)	37	−0.12 (−0.28; 0.04)	0.14	0.01 (−0.02; 0.04)	0.60	NA	NA	NA	NA
Fat mass (%)	36	0.74 (−1.81; 3.29)	0.56	−0.27 (−0.82; 0.28)	0.32	0.63 (−1.72; 2.98)	0.59	−0.14 (−0.66; 0.39)	0.60
Fat free mass (kg)	36	0.16 (−0.56; 0.88)	0.66	0.02 (−0.14; 0.17)	0.82	0.15 (−0.39; 0.68)	0.58	0.02 (−0.10; 0.14)	0.75
PWV (m/s)	28	0.15 (−0.16; 0.46)	0.34	0.02 (−0.04; 0.08)	0.50	0.14 (−0.18; 0.47)	0.37	0.02 (−0.04; 0.09)	0.44

BMI = body mass index; SBP = systolic blood pressure; DBP = diastolic blood pressure; PWV = pulse wave velocity; NA = not applicable. ^1^ Results show the association between preconception food score and offspring cardiovascular health keeping preconception MVPA at a constant level, and the association between preconception MVPA and offspring cardiovascular health keeping the preconception food score constant. Linear regression model used for analysis: Offspring cardiovascular health = preconception food score x preconception MVPA x confounders. ^2^ Adjusted for offspring sex and offspring age. We did not adjust outcomes expressed in Z-scores, as sex and age were already taken into account when calculating a Z-score.

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
