# Peer review of "Preconception Lifestyle and Cardiovascular Health in the Offspring of Overweight and Obese Women"

_nutrients, 2019, doi:10.3390/nu11102446_

Round 1

Reviewer 1 Report

Thank you for the opportunity to review this article. The manuscript is well written and nice to read. This is a very small study with an important overall aim: to understand the impact of pre-conception intervention (dietary and sport) on off-springs cardiovascular health.

Sample size is rather small and the study has a very high drop out rate. The authors present a comparison between participants and non-participants to demonstrate that the study group is representative of the whole sample. However, due to the small sample size I suggest to add a power analysis to the manuscript. 

Please add to table 1 and table S1 whether any off-spring were born preterm. Infertility treatment has an increased risk for preterm birth. If there are preterm born in the study cohort please check, whether results (especially those unexpected "sugar consumption" results) might be confounded by preterm birth.

Please discuss the clinical relevance of a DBP Z-score reduction of 0.05 / PWV reduction of 0.06 m/s.

Despite the limitations I think it important to report these data as a basis for future studies.

Author Response

Reviewer I:

Sample size is rather small and the study has a very high drop-out rate. The authors present a comparison between participants and non-participants to demonstrate that the study group is representative of the whole sample. However, due to the small sample size I suggest to add a power analysis to the manuscript. 

Response: We have added a power analysis to the manuscript (line90-95)

Please add to table 1 and table S1 whether any off-spring were born preterm. Infertility treatment has an increased risk for preterm birth. If there are preterm born in the study cohort please check, whether results (especially those unexpected "sugar consumption" results) might be confounded by preterm birth.

Response: None of the children in the follow up study was born prematurely (as indicated in table 1).

Please discuss the clinical relevance of a DBP Z-score reduction of 0.05 / PWV reduction of 0.06 m/s.

Response: We have added a discussion of the clinical relevance of the difference in lines 255 to 256.

Reviewer 2 Report

This manuscript report associations between pre-conception dietary habits and PA levels and offspring anthropometry measured between 3 to 6 years old. The original study was a RCT in infertile obese women with a 6 months lifestyle intervention. This is very interesting since few pre-conception lifestyle interventions have been conducted, especially with longer follow-up on offspring outcomes. I have a few major comments.

The first major point is: why not report the childhood outcomes simply comparing the two groups: intervention versus control? Has this been reported before? If so this should be mentioned upfront in introduction. If not, this should be reported here, even if null – this is of high interest to the field.

Abstract: please precise ethnicity of the included population

Abstract: please precise that the population included was infertile

Abstract: was the assessment of diet and PA covering the period before entree in trial or during/end of intervention?

Intro:

Please precise that the outcomes reports to the women, and at what point of the intervention/ follow-up (at 6 month? During pregnancy?) in the statement:

The six-month preconception lifestyle intervention favorably changed lifestyle [8], reduced weight [7], and improved cardiometabolic health [9]

Methods:

It is unclear to me how were the repeated measures of exposures (diet or PA) included in the models – each time point individually? Mean across all time-points measured? More advanced longitudinal modeling?

How was considered the treatment arm variable in the analyses?

Why exclude the 5 women with the highest MVPA? They may be informative if true. The authors could have analyzed in categories

Was the distribution of variables considered? Any transformation done?

Results:

In Table S1, please include the number that were initially in each treatment arm (intervention/ control) in both group (included participants and non-participants)

The measures of diet and PA should be presented for each time point that the investigators have measured in pre-conception (0, 3, 6, and 12 months), for the overall group and divided by treatment arm

The adjusted models are taking into account only child sex and age at measurements. Additional adjusted model should take into account other potential confounders, including maternal pre-pregnancy BMI, smoking, ethnicity, and education (at the minimum). The authors states that adding maternal covariables did not changes their results, but the list of covariables they mentioned does not include maternal pre-pregnancy BMI or smoking.

The analyses should take into account treatment arm: at the minimum include treatment and potential interaction term in the models (and stratified if interaction detected, but that will be difficult given the low sample size to start)

Author Response

Reviewer II:

The first major point is: why not report the childhood outcomes simply comparing the two groups: intervention versus control? Has this been reported before? If so this should be mentioned upfront in introduction. If not, this should be reported here, even if null – this is of high interest to the field.

Response: The paper describing effects of the intervention on child health is currently under review elsewhere. It does not show any effects of the intervention on child health. We now state this in the introduction (line 64-67)

Abstract: please precise ethnicity of the included population

We added details about ethnicity in the abstract (line 32)

Abstract: please precise that the population included was infertile

We have added this to line 33 in the abstract.

Abstract: was the assessment of diet and PA covering the period before entree in trial or during/end of intervention?

We have now clarified this in the abstract (line 35) diet and PA were assessed during the intervention.

Intro: Please precise that the outcomes reports to the women, and at what point of the intervention/ follow-up (at 6 month? During pregnancy?) in the statement:

The six-month preconception lifestyle intervention favorably changed lifestyle [8], reduced weight [7], and improved cardiometabolic health [9]

We have now clarified the sentence: The six month preconception lifestyle intervention favorably changed lifestyle during the intervention which led to reduced weight and improved cardiometabolic health 6 months after the start of the intervention. (line 63-64)

Methods: It is unclear to me how were the repeated measures of exposures (diet or PA) included in the models – each time point individually? Mean across all time-points measured? More advanced longitudinal modeling?

Although we assessed diet and PA repeatedly, for these analyses we used preconception diet and PA as the last measured diet and PA before conception, as described in line 108-112.

How was considered the treatment arm variable in the analyses?

This was done by adding treatment allocation as a binary variable to the regression

models. We added this to the methods section in line 173.

Why exclude the 5 women with the highest MVPA?

These women were excluded from the analyses as an average of 5 hours per day of physical activity (as was reported by these women) is very likely to be misreported.

Was the distribution of variables considered? Any transformation done?

The distribution of the covariates as well as the residuals has been assessed as reported in the methods section (line 176). No transformations were done.

Results: In Table S1, please include the number that were initially in each treatment arm (intervention/ control) in both group (included participants and non-participants)

We have added this to the table as requested.

The measures of diet and PA should be presented for each time point that the investigators have measured in pre-conception (0, 3, 6, and 12 months), for the overall group and divided by treatment arm

This information has already been reported elsewhere: it was published in PlosOne 2018 Nov 7; 13(11);e0206888

The adjusted models are taking into account only child sex and age at measurements. Additional adjusted model should take into account other potential confounders, including maternal pre-pregnancy BMI, smoking, ethnicity, and education (at the minimum). The authors states that adding maternal covariables did not changes their results, but the list of covariables they mentioned does not include maternal pre-pregnancy BMI or smoking.

Also adjusting for prepregnancy BMI and smoking did not alter the results. We have added this to the manuscript (line 250)

The analyses should take into account treatment arm: at the minimum include treatment and potential interaction term in the models (and stratified if interaction detected, but that will be difficult given the low sample size to start)

As described in the methods, we have additionally adjusted for treatment (see line 172). Considering the small number of observations and the large number of tests, we chose not to asses additional interaction terms with treatment allocation.

Round 2

Reviewer 1 Report

My concerns have been addressed. I have no further comments.

Author Response

We are happy to read that the reviewer is satisfied with our paper.

Reviewer 2 Report

the authors have addressed my previous comments

Author Response

(The authors gave the same response as above.)
